# Sex differences in rates of permanent pacemaker implantation and in-hospital complications: A statewide cohort study of over 7 million persons from 2009–2018

**Vijayatubini Vijayarajan** [ORCID]**, Leonard Kritharides, David Brieger, Yeu-Yao Cheng, Vincent Chow, Austin Chin Chwan Ng** [ORCID]*

Department of Cardiology, Concord Hospital, The University of Sydney, Concord, NSW, Australia

* chin.ng@sydney.edu.au

**Data Availability Statement:** The New South Wales Population and Health Services Research Ethics Committee prohibits the authors from

## Abstract

### Background

Whether a bias exists in the implantation of permanent pacemakers (PPI) and complications according to sex and age in the Australian population is unclear.

### Hypothesis

Population rate of PPI and its complications differed between men and women.

### Methods

We examined the prevalence of PPI from January-2009 to December-2018 from datasets held by the New South Wales (NSW) Centre-for-Health-Record-Linkage, including patient's characteristics and in-hospital complications. All analysis was stratified by sex and age by decade.

### Results

A total of 28,714 admissions involved PPI (40% women). The mean PPI rate (±standard-deviation) and median age (interquartile range) was 2,871±242 per-annum and 80yrs (73-86yrs), respectively. At the same time-period, the mean NSW population size was 7,487,393±315,505 persons (50% women; n = 3,773,756±334,912). The mean annual age-adjusted rate of PPI was 125.5±11.6 per-100,000-men, compared to 63.4±14.3 per-100,000-women (P<0.01). The mean annual rate of PPI increased from 2009–2017 by 0.9 ±3.3% in men, compared to 0.4±4.4% in women (P<0.01) suggesting a widening disparity. Total non-fatal in-hospital complications was higher in women compared to men (8.2% vs 6.6%, P<0.01), and this persisted throughout study period even after adjusting for multiple covariates. Overall, in-hospital mortality was low (0.73%) and similar between sexes.

making the minimal data set publicly available.
Interested researchers may contact the Ethics
coordinator (ethics@cancerinstitute.org.au) to seek
permission to access the data; data will then be
made available upon request to all interested
researchers who receive approval from the New
South Wales Population and Health Services
Research Ethics Committee.

**Funding:** The author (s) received no specific
funding for this work.

**Competing interests:** The authors have declared
that no competing interests exist

## Conclusion

In a statewide Australian population exceeding 7 million, PPI rates were consistently nearly two-fold higher for men compared to women over 10-years, with an apparently widening disparity, that was not explained by age. Overall complication rates were higher in women. Future studies should examine the aetiology behind this disparity in PPI rates, as well as its complications.

## Introduction

Cardiovascular disease remains a major cause of morbidity and mortality worldwide [1, 2]. Moreover, there are sex differences in the management of patients presenting with cardiovascular diseases [1]. In acute coronary syndromes (ACS) for example, men receive more standard of care treatment and invasive procedures than do women [1].

Pacemaker technology and utilisation has grown exponentially [3, 4]. Indications for pacemaker use are well-established [5]. There can, however, be complications associated with the PPI [3, 6], with complication rates varying between 1–6% [3] depending on the type of device implanted, access sites, and implanter's experience [7]. In addition, women have been reported to have more complications associated with PPI [8]. However, sex differences in complication rates in unselected population-level cohorts is less well studied, or whether there are sex differences in rates of PPI in general.

The primary aims of this study were to investigate sex differences in pacemaker utilisation and complications at a population-level. This was done by investigating: 1) temporal trends in PPI case-volumes and rates stratified by sex and age adjusting for changes in population size; and, 2) temporal trends in in-hospital mortality and morbidity associated with PPI stratified by sex.

## Methods

### Study population

For this study, we utilised the databases held by the Centre-for-Health-Record-Linkage (CHeReL). This facility holds one of the largest data-linkage systems in Australia, linking health data of residents living in Australia's largest state of New South Wales (NSW) [9]. From its Admitted -Patient-Data-Collection (APDC) registry, which encompasses ≥97% of all NSW healthcare facilities, we identified consecutive admissions that included a PPI (primary or secondary procedure) coded as 38353–00 under the Australian-Classification-of-Health-Interventions (ACHI) coding system between 1-January-2009 and 31-December-2018. Patients who had implantable cardioverter defibrillator or cardiac resynchronization therapy pacemaker were not considered for the purpose of this study.

### Data sources

Variables obtained from the APDC registry for each hospital admission that involved a PPI procedure include time/date of admission, age, sex, referral source, type of facility, length of admission, and whether the patient died in-hospital.

The primary and all secondary diagnoses recorded for each admission were also retrieved. Each diagnosis was coded according to the International-Classification-of-Diseases, Tenth-Revision Australian-Modification (ICD-10AM). For this study, we pre-specified the

indications for pacemaker by identifying specific ICD-10AM codes listed either as primary or secondary diagnosis under the following categories: 1) complete heart block (CHB); 2) other atria-ventricular (AV) block and bradycardia; 3) sick sinus syndrome (SSS); and/or, 4) others (see S1 Table for ICD-10AM codes). If more than one indication was coded, each was recorded. We separately identified whether ACS was listed as primary diagnosis for admission, and if concomitant cardiac procedures, including coronary-artery-bypass-graft (CABG) surgery or on cardiac valves were performed during admission. Additional comorbidities of interest extracted for this study are presented as baseline characteristics in Table 1 (see S1 Table for each comorbidity ICD-10AM codes). We also semi-quantified the overall comorbid status of each patient based on the Charlson comorbidity index (CCI) [10, 11].

## Study outcomes

The co-primary outcomes studied were rates of PPI at a population-level and in-hospital complications. Cases were limited to only NSW residents to minimize incomplete tracking. For in-hospital complications associated with PPI, we examined rates of deep vein thrombosis (DVT), pulmonary embolism (PE), traumatic cardiac injuries (pericardial effusion and cardiac tamponade), infection post device insertion, pneumothorax, haemothorax, mechanical complications, lead and generator manipulations, and others (including embolisms/fibrosis/haemorrhage/pain/stenosis/thrombosis) (see S1 Table for each complication definition). We also examined in-hospital cause-specific mortality based on published classifications [12]. All deaths were coded independently by two reviewers (AN/VV) according to general principles set by the World Health Organization [13], with disparities resolved by a third reviewer (VC). Reviewers were blinded to patient's comorbidities during coding.

The NSW Population and Health Services Research Ethics Committee granted a waiver of the usual requirement for the consent of the individual to the use of their health information (reference number: 2013/09/479). All patient data were de-identified and analyzed anonymously.

## Statistical analysis

All admissions involving NSW residents between 1-January-2009 and 31-December-2018 were initially collected. The study cohort was then limited to NSW residents, stratified by sex, and confined to de novo PPI to reduce confounders. For this study, *de novo* PPI is defined as patients who had a single PPI procedure during index admission (excluding those who had recurrent procedures or generator replacement between 1-January-2009 and 31-December-2018). This cohort was used to determine the incidence rate and temporal trend of PPI procedures. The age-adjusted rate of implantation for a specific calendar-year was calculated by dividing the number of PPI admissions by age categories in decade-year age groups for that specific year over the state population size in corresponding age groups of that year, stratified by sex. The NSW population characteristics for each calendar-year from 2009 to 2018 was obtained from publicly available resources held by the Australian Bureau of Statistics [14]. The same cohort was used to analyse the non-fatal complication rates of PPI. A separate analysis was performed to examine the in-hospital non-fatal complication rates of PPI based on all admission cases rather than only on *de novo* PPI.

All continuous variables were expressed as median (IQR, interquartile range) unless specified otherwise, with absolute values in numbers and proportions in percentages. To compare categorical variables, Fisher exact and chi-square tests were used whereas Mann-Whitney U test was used to compare continuous variables. Simple linear regression was used to assess

**Table 1. Baseline characteristics.**

| Parameters | Total cohort (n = 28,714) | Stratified by sex | | P value |
|---|---|---|---|---|
| | | Male (n = 17,261; 60.1%) | Female (n = 11,453; 39.9%) | |
| Median age (IQR) | 80 (73–86) | 79 (72–85) | 81 (75–86) | <0.001 |
| Referral source | | | | |
| Emergency department | 7,745 (27.0) | 4,371 (25.3) | 3,374 (29.5) | <0.001 |
| Elective | 13,303 (46.3) | 8,443 (48.9) | 4,860 (42.4) | |
| Inter-hospital referred | 7,274 (25.3) | 4,205 (24.4) | 3,069 (26.8) | |
| Others | 285 (1.0) | 179 (1.0) | 106 (0.9) | |
| Unknown | 107 (0.4) | 63 (0.4) | 44 (0.4) | |
| Type of facility | | | | |
| Public | 14,850 (51.7) | 8,700 (50.4) | 6,150 (53.7) | <0.001 |
| Private | 13,864 (48.3) | 8,561 (49.6) | 5,303 (46.3) | |
| Indication for PPM* | | | | |
| Sick sinus syndrome | 7,822 (27.2) | 3,919 (22.7) | 3,903 (34.1) | <0.001 |
| Complete heart block | 5,301 (18.5) | 3,355 (19.4) | 1,946 (17.0) | <0.001 |
| Other AV block and bradycardia | 12,261 (42.7) | 7,949 (46.1) | 4,312 (37.6) | <0.001 |
| Others† | 5,603 (19.5) | 3,396 (19.7) | 2,207 (19.3) | 0.40 |
| Other primary diagnosis | | | | |
| Acute coronary syndrome | 603 (2.1) | 380 (2.2) | 223 (1.9) | 0.15 |
| Concomitant cardiac procedures during admission | | | | |
| CABG | 572 (2.0) | 423 (2.5) | 149 (1.3) | <0.001 |
| All cardiac valves surgery | 981 (3.4) | 583 (3.4) | 398 (3.5) | 0.67 |
| TAVI | 113 (0.4) | 63 (0.4) | 50 (0.4) | 0.39 |
| Comorbidities | | | | |
| Cardiovascular diseases | | | | |
| Congestive cardiac failure | 2,261 (7.9) | 1,255 (7.3) | 1,006 (8.8) | <0.001 |
| Ischaemic heart disease | 3,679 (12.8) | 2,462 (14.3) | 1,217 (10.6) | <0.001 |
| Previous PCI / CABG | 2,273 (7.9) | 1,776 (10.3) | 497 (4.3) | <0.001 |
| Atrial fibrillation | 6,800 (23.7) | 3,781 (21.9) | 3,019 (26.4) | <0.001 |
| Peripheral vascular disease | 737 (2.6) | 520 (3.0) | 217 (1.9) | <0.001 |
| Valvular heart disease | 1,164 (4.1) | 702 (4.1) | 462 (4.0) | 0.90 |
| Prosthetic heart valves | 532 (1.9) | 352 (2.0) | 180 (1.6) | 0.004 |
| Previous strokes | 317 (1.1) | 178 (1.0) | 139 (1.2) | 0.08 |
| Cardiac risk factors | | | | |
| Hypertension | 5,819 (20.3) | 3,218 (18.6) | 2,601 (22.7) | <0.001 |
| Diabetes | 5,519 (19.2) | 3,554 (20.6) | 1,965 (17.2) | <0.001 |
| Current/ex-smoker | 8,362 (29.1) | 6,391 (37.0) | 1,971 (17.2) | <0.001 |
| Hyperlipidaemia | 517 (1.8) | 327 (1.9) | 190 (1.7) | 0.15 |
| Chronic renal failure | 1,992 (6.9) | 1,215 (7.0) | 777 (6.8) | 0.42 |
| Malignancy | 216 (0.8) | 143 (0.8) | 73 (0.6) | 0.07 |
| Charlson comorbidity index‡ | | | | |
| Mean ± SD | 0.7 ± 1.5 | 0.8 ± 1.5 | 0.7 ± 1.4 | |
| Median (IQR) | 0 (0–1) | 0 (0–1) | 0 (0–1) | <0.001 |
| Length of stay, days | | | | |
| Median (IQR) | 3 (1–7) | 2 (1–7) | 3 (1–8) | <0.001 |
| Length of stay based on referral source, days | | | | |
| Median (IQR) | | | | |
| Emergency | 8 (4–13) | 7 (4–12) | 8 (5–14) | <0.001 |

*(Continued)*

**Table 1.** (Continued)

| Parameters | Total cohort (n = 28,714) | Stratified by sex | | P value |
|---|---|---|---|---|
| | | Male (n = 17,261; 60.1%) | Female (n = 11,453; 39.9%) | |
| Elective | 1 (1–3) | 1 (1–2) | 1 (1–3) | <0.001 |
| Inter-hospital referred | 3 (2–6) | 3 (2–6) | 4 (2–7) | <0.001 |
| Others | 1 (1–6) | 1 (1–5) | 1 (1–7) | 0.21 |
| Unknown | 3 (1–8) | 2 (1–8) | 6 (1–9) | 0.31 |

Values represent number of patients with values in brackets representing percentages, or otherwise stated.

CABG, coronary artery bypass graft; IQR, interquartile range; PCI, percutaneous coronary intervention; PPM, permanent pacemaker; TAVI, transcutaneous aortic valve implantation; yo, years old.

* If more than one prespecified indication for PPM was coded during admission (see Methods for the indications for PPM), each was recorded; thus, a patient may have more than one indication for PPM coded during admission.

† Pacemaker implanted for tachybrady arrhythmic syndrome or unexplained syncope.

‡ Conditions included in the Charlson Comorbidity Index include myocardial infarction, congestive cardiac failure, peripheral vascular disease, stroke, dementia, chronic pulmonary disease, connective tissue disease, peptic ulcer disease, liver disease (mild vs. moderate to severe), diabetes (with or without organ damage), hemiplegia, moderate to severe renal disease, any tumor (within last 5 years), lymphoma, leukemia, metastatic solid tumor and acquired immunodeficiency syndrome (AIDS).

temporal trends of cases and events during the study period. Binary logistic regression was used to determine independent predictors for total in-hospital non-fatal complications, and separately for in-hospital death. Considered univariables include age, sex, referral source, facility type, indications for PPI, ACS, concomitant cardiac procedures, calendar-year of PPI, and comorbidities. A tolerance of >0.4, equating to a variance inflation factor >2.5 was set to avoid any potential multicollinearity. All analyses were performed using SPSS-v23 (IBM-USA). A P-value <0.05 was considered statistically significant.

## Results

Between 1-January-2009 and 31-December-2018, there were 28,714 *de novo* PPI after excluding non-NSW residents (n = 414) and recurrent admissions identified as an admission involving PPI (n = 7,033) (S1 Fig).

### Incidence rate and temporal trend of PPI caseload

During the 10-year study period, the mean (±SD) implantation rate was 2,871±242 cases per-annum. Though the annual volume of PPI during the study period was steady for both sexes, more men had PPI (Fig 1A). The mean total statewide population was 7,487,393±315,505 persons during the study period, with 50% women (n = 3,773,756±334,912). The age-adjusted mean annual PPI was 125.5±11.6 per-100,000-men compared to 63.4±14.3 PPI per-100,000-women, P<0.01 (Fig 1B). Total volume of PPI increased exponentially beyond 40-49yo age group in both sexes (S2 Fig). When stratified by age groups above 50yo, the rate of PPI per-100,000 in men was consistently double that of women throughout the study period, with the disparity less pronounced in age groups below 50yo (S3 and S4 Figs). From 2009–2017 (excluding 2018 to limit ascertainment bias), the mean annual rate of increase in PPI in men was 0.9±3.3% compared to 0.4±4.4% in women (P<0.01), suggesting a widening disparity.

### Baseline characteristics

There were 28,714 *de novo* PPI from 2009–2018 (39.9% women [n = 11,453]) (Table 1). The study cohort's median (IQR) age was 80yo (73-86yo). Women were more often referred from

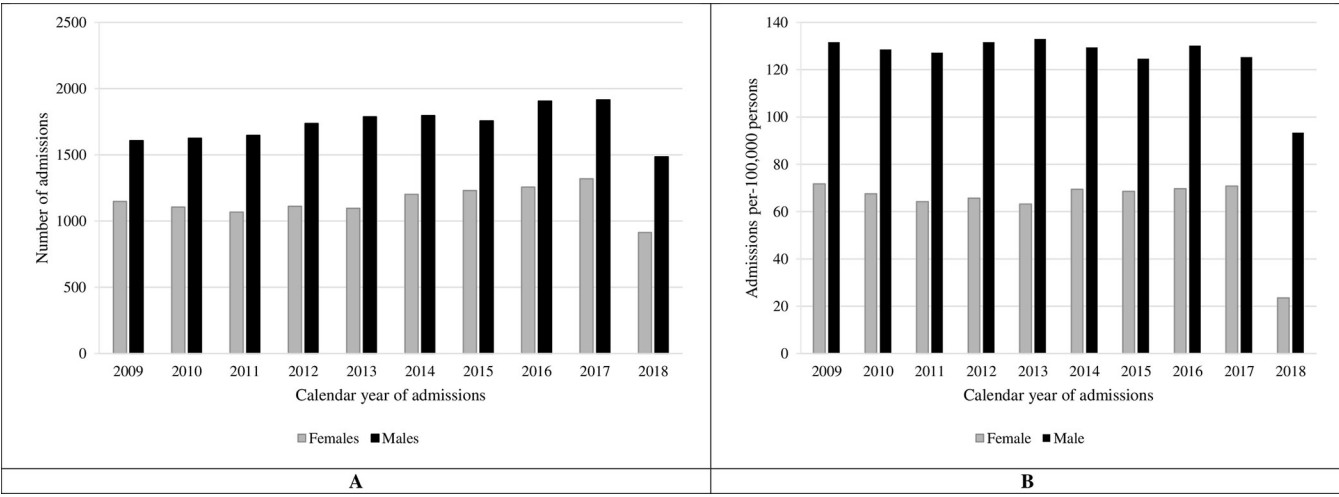

**Fig 1. Total permanent pacemaker implantation admissions based on calendar-year and stratified by sex.** A. Shows the total number of permanent pacemaker implantation admissions per calendar-year, stratified by male (black bar) (2009–2018: n = 17,261, linear regression for trend P = 0.38; 2009–2017: n = 15,776, linear regression for trend P<0.001) and female (grey bar) (2009–2018: n = 11,453, linear regression for trend P = 0.73; 2009–2017: n = 11,453, linear regression for trend P<0.01). B. Shows the age-adjusted permanent pacemaker implantation admission rates per-100,000-persons per calendar-year, stratified by male (black bar) (2009–2018: linear regression for trend P = 0.07; 2009–2017: linear regression for trend P = 0.25) and female (grey bar) (2009–2018: linear regression for trend P = 0.17; 2009–2017: linear regression for trend P = 0.54). Mean annual age-adjusted (by decade) rate of PPI was 125.5±93.4 per-100,000-males, compared to 63.4±14.3 per-100,000-females (P<0.01) during study period.

the Emergency Department or were inter-hospital transfers. The indications for PPI also differed between sexes, with women receiving PPI more often for SSS while men had a higher rate of CHB, AV block or bradycardia. Atrial fibrillation (AF) was more common in women. Median CCI score was 0 (0–1). 2.3% of patients had a primary diagnosis of ACS. Overall, concomitant cardiac procedures during the admission for PPI were few.

### In-hospital complications during PPI admission

The rate of total non-fatal complications was 7.2% (n = 2,077) and was higher in women (8.2% vs 6.6% in men, P<0.001) (Table 2). Venous thromboembolism was recorded in 44 patients (8 PEs and 38 DVTs), with no difference between sexes. Overall, total non-fatal complications remained steady during the study period based on linear regression trend analysis. (Fig 2). In a separate analysis that assessed non-fatal complication rates based on all PPI admissions (n = 35,747 cases), though the absolute number for each category of complications was higher, the derived rates did not differ significantly from that observed for the *de novo* study cohort (S2 Table).

In multivariable analysis, men had a lower risk of total in-hospital non-fatal PPI complications (adjusted odds ratio (aOR) 0.79, 95% confidence interval (CI) 0.72–0.87, P<0.001) (Table 3; see S3 Table for univariable analysis). In addition, the final two years (2017 and 2018) were independently associated with a lower risk compared to reference year-2009. Independent predictors of increased risk include CHB, ACS presentation, concomitant cardiac procedures, history of valvular heart disease, AF, and hypertension.

We further investigated the predictors of in-hospital non-fatal complications in women and separately for men (S7–12 Tables). Similar independent predictors for increased in-hospital complications risk in both men and women were younger age, cardiac valve surgery including TAVI, valvular heart disease, atrial fibrillation and malignancy, while referred as an elective procedure was associated with decreased risk of complications for both genders. Independent predictors associated with increased risk for complications in women but not in men

**Table 2. In-hospital complications during permanent pacemaker implantation*.**

| Complications, no. (%) | Total cohort (n = 28,714) | Stratified by sex | | P value |
|---|---|---|---|---|
| | | Male (n = 17,261; 60.1%) | Female (n = 11,453; 39.9%) | |
| Total non-fatal complications | 2,077 (7.23) | 1,142 (6.62) | 935 (8.16) | <0.001 |
| Venous thromboembolism | 44 (0.15) | 22 (0.13) | 22 (0.19) | 0.22 |
| Pulmonary embolism | 8 (0.02) | 3 (0.02) | 5 (0.04) | 0.28 |
| Deep venous thrombosis | 38 (0.13) | 20 (0.12) | 18 (0.16) | 0.41 |
| Infection post-implantation | 177 (0.62) | 106 (0.61) | 71 (0.62) | 0.94 |
| Pocket complications | 7 (0.02) | 5 (0.03) | 2 (0.02) | 0.71 |
| Cardiac injuries | 37 (0.13) | 12 (0.07) | 25 (0.22) | <0.01 |
| Pericardial effusion | 173 (0.60) | 81 (0.47) | 92 (0.80) | <0.01 |
| Cardiac tamponade | 54 (0.19) | 22 (0.13) | 32 (0.28) | <0.01 |
| Pneumothorax | 322 (1.12) | 148 (0.86) | 174 (1.52) | <0.01 |
| Haemothorax | 0 (0) | 0 (0) | 0 (0) | NA |
| Mechanical complications† | 462 (1.61) | 266 (1.54) | 196 (1.71) | 0.27 |
| Lead manipulation | 368 (1.28) | 217 (1.26) | 151 (1.32) | 0.71 |
| Generator manipulation | 21 (0.07) | 14 (0.08) | 7 (0.06) | 0.88 |
| Others‡ | 642 (2.23) | 361 (2.09) | 281 (2.45) | 0.05 |
| In-hospital mortality | 209 (0.73) | 120 (0.69) | 89 (0.78) | 0.44 |

Values represent number of patients with values in brackets representing percentages, or otherwise stated.

NA, not applicable.

* Based on single episode of permanent pacemaker implantation during index admission

† Defined as breakdown, displacement, malposition, leakage, obstruction, perforation or protrusion.

‡ Include embolisms, fibrosis, haemorrhage, pain, stenosis or thrombosis.

include complete heart block presentation, history of peripheral vascular disease and hypertension. In comparison, these characteristics were associated with increased risk for complications in men but not observed in women: concomitant CABG during PPI admission, history of ischaemic heart disease, congestive cardiac failure, stroke, chronic pulmonary disease and kidney disease.

A total of 209 (0.73%) patients died in-hospital, with no significant difference between sexes (Table 2). In-hospital mortality was not dependent on sex, nor was there significant change in mortality during the study period (S4 Table). Overall, cardiovascular causes of death (51.2%) were more common than noncardiovascular causes (48.8%), with heart failure the most common in-hospital cause-specific death (22.0%) during PPI admission (S5 Table). Sepsis accounted for 18.2% of total in-hospital deaths, while PE accounted for only 2.9% of deaths. Causes of deaths did not differ significantly between sexes (p = 0.46). There were 3 deaths directly attributed to the PPI: 2 from direct cardiac injury (lacerated coronary sinus and perforated right ventricle) and 1 from pacemaker site infection.

## Discussion

The present study examined the epidemiology and in-hospital complications of PPI in an unselected statewide Australian population over a 10-year period. We identified the following key points: 1) men consistently received more PPI even after adjustment for age; 2) rates of PPI appeared to be increasing regardless of sex; 3) women suffered higher rates of in-hospital PPI complications; and 4) in-hospital mortality was low, with no difference between sexes, and no significant improvement during the study period.

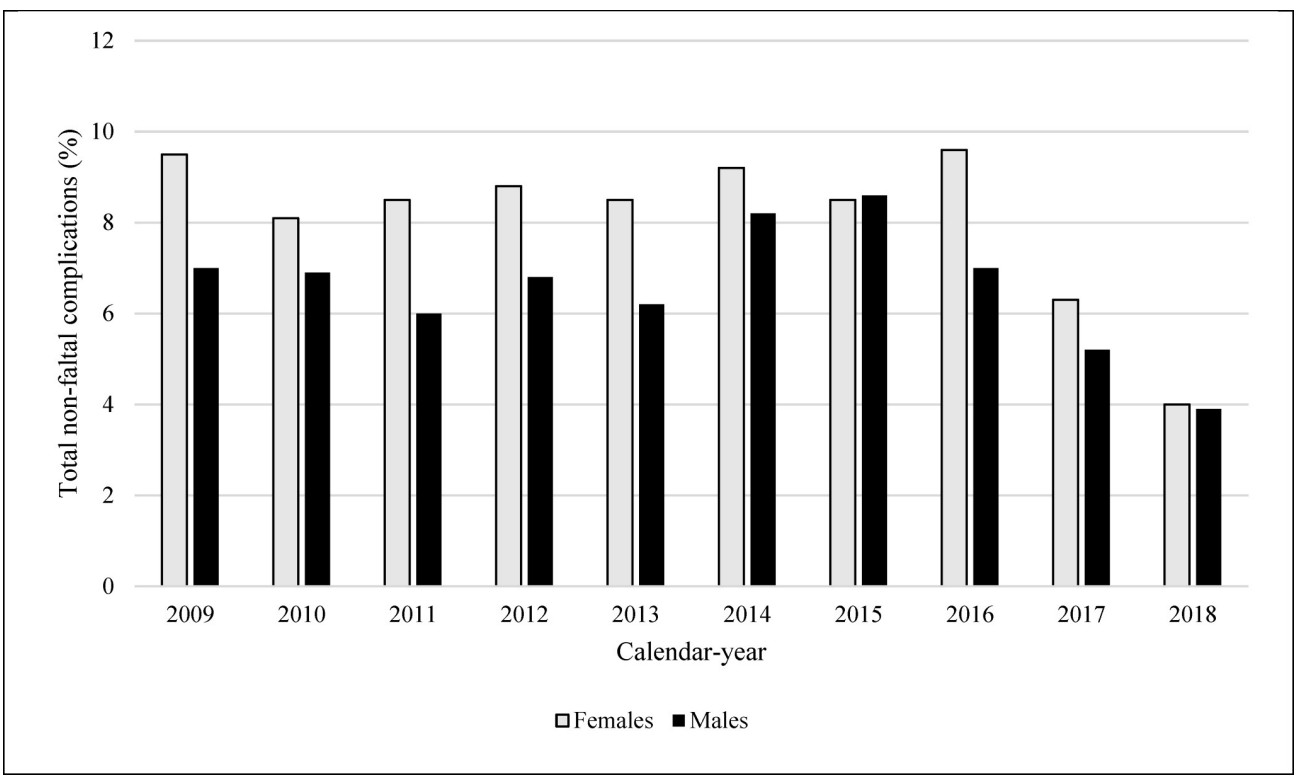

**Fig 2. Temporal trend of rates of total non-fatal complications during permanent pacemaker implantation admissions.** Fig 2 shows the rates of in-hospital total non-fatal complications during admission for permanent pacemaker implantation stratified by calendar-year, and males (black bar) (2009–2018: n = 1,142, linear regression for trend P = 0.30; 2009–2017: n = 1,084, linear regression for trend P = 0.97) versus females (grey bar) (2009–2018: n = 935, linear regression for trend p = 0.06; 2009–2017: n = 898, linear regression for trend P = 0.33).

### Sex differences in PPI utilisation

The median age of women were about 2 years older at time of PPI compared to men in this study. Sick sinus syndrome occurs primarily in older adults [15], and we noted women had a higher prevalence of sick sinus syndrome as an indication for PPI, which could partially account for the observation of women receiving a pacemaker at an older age. In addition, men are known to have a lower life expectancy (5 years earlier) compared to women [2], and as shown in this present study, they also have higher prevalence of ischaemic heart disease, complete heart block and other AV block and bradycardia arrhythmias. The combination of lower life expectancy and multiple cardiovascular comorbidities could account for men receiving PPI at a younger age.

We identified a nearly two-fold difference in the rate of PPI between men and women which persisted over a 10-year period implying potential unexplained systematic and sustained factors underlying this discrepancy. We are unable to comment on whether differences in the community prevalence of indications for PPI (such as SSS versus AV block) could explain this difference, but this seems unlikely given prior literature. There is a greater proportion of women receiving PPI in more urgent situations (emergency and inter-hospital transfer). This may suggest that it is the planned, elective PPI that are systematically underperformed in women in NSW. Although it is reported that women often present with atypical symptoms that may contribute to delay in diagnosis and subsequent referral for invasive procedures in the setting of coronary disease [22], there is no literature that we are aware of indicating that syncope is differentially reported between sexes, and this is unlikely to explain our findings.

**Table 3. Independent predictors for total in-hospital non-fatal complications*.**

| Parameters | Odds ratio (95% CI) | P value |
|---|---|---|
| Male | 0.79 (0.72–0.87) | <0.001 |
| Year of admission | | <0.001 |
| 2009 | 1.00 (reference) | |
| 2010 | 0.92 (0.75–1.13) | 0.42 |
| 2011 | 0.90 (0.73–1.10) | 0.31 |
| 2012 | 0.93 (0.76–1.13) | 0.47 |
| 2013 | 0.85 (0.69–1.04) | 0.11 |
| 2014 | 1.01 (0.83–1.22) | 0.94 |
| 2015 | 1.02 (0.84–1.24) | 1.02 |
| 2016 | 0.99 (0.82–1.21) | 0.95 |
| 2017 | 0.65 (0.52–0.79) | <0.001 |
| 2018 | 0.40 (0.31–0.51) | <0.001 |
| Age–per 1-year increase | 0.989 (0.985–0.993) | <0.001 |
| Referral source | | <0.001 |
| Emergency department | 1.00 (reference) | |
| Elective | 0.62 (0.55–0.71) | <0.001 |
| External hospital-referred | 0.79 (0.70–0.89) | <0.001 |
| Others | 0.64 (0.38–1.07) | 0.09 |
| Unknown | 0.83 (0.43–1.60) | 0.58 |
| Type of facility | | |
| Public | 1.00 (reference) | |
| Private | 0.78 (0.70–0.87) | <0.001 |
| Complete heart block | 1.22 (1.08–1.36) | <0.001 |
| Sick sinus syndrome | 0.91 (0.82–1.02) | 0.12 |
| Acute coronary syndrome | 1.36 (1.06–1.75) | 0.02 |
| CABG | 1.41 (1.10–1.81) | 0.01 |
| All cardiac valve surgery | 2.88 (2.37–3.50) | <0.001 |
| TAVI | 5.49 (3.46–8.71) | <0.001 |
| Valvular heart disease | 1.55 (1.30–1.86) | <0.001 |
| Atrial fibrillation/flutter | 1.34 (1.21–1.49) | <0.001 |
| Hypertension | 1.16 (1.04–1.30) | 0.01 |
| CCI score–per 1-score † | 1.10 (1.07–1.13) | <0.001 |

CABG, coronary artery bypass graft; TAVI, transcutaneous aortic valve implantation; CCI, Charlson comorbidity index; CI, confidence interval

* Multivariable binary logistic regression method was used to identify independent predictors for all in-hospital complications; only univariables with P<0.05 were included in the multivariable analysis (refer to Supplementary Table 3 for univariable analysis).

† Conditions included in the Charlson Comorbidity Index include myocardial infarction, congestive cardiac failure, peripheral vascular disease, stroke, dementia, chronic pulmonary disease, connective tissue disease, peptic ulcer disease, liver disease (mild vs. moderate to severe), diabetes (with or without organ damage), hemiplegia, moderate to severe renal disease, any tumor (within last 5 years), lymphoma, leukemia, metastatic solid tumor and acquired immunodeficiency syndrome (AIDS)

Other possibilities include physician avoidance of PPI in women because of systematic bias, or concern regarding a perceived increased risk of complications in women, multiple comorbidities and smaller body size presenting a more challenging implantation [17]. As noted in the present study, we confirmed that women were at greater risk of non-fatal complications during

PPI compared to men, but in absolute terms the risks are small and would not justify such discrepant implantation rates.

There is conflicting evidence in the literature regarding PPI rates between sexes [16–20]. Boccia et al and Chen et al described no sex difference in PPI utilization in an Italian study and a study on 7,203 PPI in China, respectively [19, 20]. In contrast, Uslan et al found PPI rates were greater in men in Olmsted County, Minnesota [17]. Eccleston et al also observed a higher incidence of PPI in men, though the study was limited to only 14 private Australian hospitals [8]. Westaway et al reported higher rates of PPI in men in an Australian population. However, their study did not provide additional baseline characteristics including comorbidities of the study cohort, nor was their complication rates linked to the study cohort [21]. Moore et al described that more women received pacemakers compared to men but their study cohort encompassed all cardiac implantable electronic devices (CIED) implantations in Australia confined to a five-year study period. This study did not analyse the trend and the number of PPI was not age-adjusted for the local population size [22]. In contrast, the present study focused on pacemakers in an unselected statewide cohort spanning 10-years. We found men received more pacemakers compared to women despite adjusting for population size and age.

Epidemiological studies on incidence of cardiac conduction disease remain few in the literature. Jensen et al carried out a prospective, population-based study in the United States (US) and found no difference in incidence of SSS between sexes [15]. In contrast, Manolio et al described a sex difference in incidence of conduction disease in 5,201 US adults above 65yo [23]. Bradycardia or conduction blocks overall were higher in men (5.6% vs 1.9%). Shan et al described higher rates of CHB in men (0.05% vs 0.03%) in a study of 15 million people in China [24], as did Kojic et al in a prospective study of 18,912 residents of Reykjavik [25]. However there is no convincing evidence that age-dependent conduction disease requiring PPI in women is half that in men and therefore this cannot fully explain our findings.

The increase in PPI over time observed in the present study is consistent with other population-based studies [17, 18]. Uslan et al described similar trends, with PPI rate increasing from 40 to 120 per-100000-person-years in men and 30 to 90 per-100000-person-years in females from 1975–2004 [17]. In the present study, we found the growth rate of PPI in men was double that of women, suggesting that the bias in implantation rates may be increasing.

## Sex differences in rates of in-hospital PPI complications and in-hospital mortality

The observed complications rates related to PPI appear comparable to other local and international studies [16, 22]. Despite adjusting for multiple variables, complications rates were still higher in women and this is consistent with other studies [22, 26]. Our study demonstrates the trend in overall complication rates was at least stable, with the rates in the final 2 years of the study period (2017 and 2018) being respectively 35% and 60% lower than reference year-2009. Findings from year-2018 should be treated cautiously as it is does not consider the full cohort of that year. Future studies should seek to verify if this improving trend will continue.

Total in-hospital mortality in the present study was low at 0.7%, with no significant difference between sexes, and is consistent with that of Moore et al who reported a mortality of 0.6% [22]. In-hospital mortality in our study did not alter over the 10-year period.

## Clinical implications and future directions

Despite relatively stable rates of PPI, the absolute volume of PPI has increased due to both increasing population size and an ageing population. However, the surprising finding in the present study was despite advances in technology, and greater utilisation, there was

consistently higher rates of PPI in men than in women in the Australian population and the gap appears widening. Prospective evaluation on the use of PPI as a function of community prevalence of indications for PPI will be required.

## Limitations

This study is limited by being retrospective. NSW APDC is an administrative database, thus comprehensive clinical data such as symptoms, echocardiogram parameters and medications are not available. One of the major limitations of this study was unknown sex-based rates of bradyarrhythmia and conduction disease in our statewide population that required PPI, which limits our interpretation of whether a true gender bias exists or was driven by differential rates of conduction disease between gender that requires PPI. We found scarce data available in the literature reporting on the prevalence of bradyarrhythmia at a population-level, and we consider this to be an important area for future research. Non-fatal clinical outcomes and indication for pacemakers were determined according to ICD coding, which may be subject to ascertainment bias. However, our reported complications rates were comparable to published literature, providing certain validity to our findings [22, 26]. We also did not have clinical data such as body mass index, proceduralist experience, whether prophylactic antibiotic was used, or the rates of prophylaxis measures against venous thromboembolism, which could all potentially act as confounders on the relationship between sex and outcomes measured in the present study. We were not able to differentiate whether single or dual chamber pacemakers were implanted from this dataset. This study did not analyse long-term outcomes after PPI. There is incomplete data collection during the final year of the study for 2018, and findings for this year should be interpreted with caution. However, the large cohort coupled with a reasonably long study period allowed temporal trend analysis of not only caseload over time but outcomes as well.

## Conclusion

In this statewide population study of over 7 million persons, overall PPI increased over a 10-year period, with a significant disparity in the rates of implantation between sexes despite adjustment for population size and age. Women experienced more complications than men.

## Supporting information

**S1 Fig. Study cohort derivation.** Flow chart shows the derivation of the study cohort. Abbreviations: APDC, Admitted Patient Data Collection; CHeReL, Centre for Health Record Linkage; NSW, New South Wales State of Australia; PPM, permanent pacemaker.
(TIFF)

**S2 Fig. Total number of permanent pacemaker implantation admissions based on age groups stratified by sex.** S2 Fig shows the total number of permanent pacemaker implantation admissions during study period (2009–2018) in age groups, stratified by sex. The thick line represents males, while the dotted line represents females.
(TIFF)

**S3 Fig. Temporal trend of permanent pacemaker implantation stratified by age groups above 50yo and sex.** S3A Fig shows the pacemaker implantation admissions per calendar-year in males stratified by age groups above 50 years old (yo). S3B Fig shows the pacemaker implantation admissions per calendar-year in females stratified by age groups above 50yo.
(TIFF)

**S4 Fig. Temporal trend of permanent pacemaker implantation stratified by age groups below 50yo and sex.** S4A Fig shows the pacemaker implantation admissions per calendar-year in males stratified by age groups below 50 years old (yo). S4B Fig shows the pacemaker implantation admissions per calendar-year in females stratified by age groups below 50yo.
(TIFF)

**S1 Table. Study comorbidities and complications International Classification of Diseases Tenth Revision Australian Modification (ICD-10AM) codes and Australian Classification of Health Interventions (ACHI) procedural codes.**
(DOCX)

**S2 Table. Permanent pacemaker implantation in-hospital complications stratified by sex (based on all admissions involving permanent pacemaker implantations during study period).**
(DOCX)

**S3 Table. Univariable predictors for total in-hospital non-fatal complications.**
(DOCX)

**S4 Table. Predictors for in-hospital death.**
(DOCX)

**S5 Table. In-hospital cause-specific death during admission for permanent pacemaker implantation.**
(DOCX)

**S6 Table. Independent predictors for total in-hospital non-fatal complications (multivariable model not including CCI).**
(DOCX)

**S7 Table. Univariable predictors for total in-hospital non-fatal complications in women.**
(DOCX)

**S8 Table. Independent predictors for total in-hospital non-fatal complications (multivariable model including CCI)*** in women.**
(DOCX)

**S9 Table. Independent predictors for total in-hospital non-fatal complications (multivariable model not including CCI)*** in women.**
(DOCX)

**S10 Table. Univariable predictors for total in-hospital non-fatal complications in men.**
(DOCX)

**S11 Table. Independent predictors for total in-hospital non-fatal complications (multivariable model including CCI)*** in men.**
(DOCX)

**S12 Table. Independent predictors for total in-hospital non-fatal complications (multivariable model not including CCI)*** in men.**
(DOCX)

## Author Contributions

**Conceptualization:** Vijayatubini Vijayarajan, Leonard Kritharides, Austin Chin Chwan Ng.

**Data curation:** Vijayatubini Vijayarajan, Austin Chin Chwan Ng.

**Formal analysis:** Vijayatubini Vijayarajan, Austin Chin Chwan Ng.

**Investigation:** Vijayatubini Vijayarajan.

**Supervision:** Leonard Kritharides, Austin Chin Chwan Ng.

**Writing – original draft:** Vijayatubini Vijayarajan.

**Writing – review & editing:** Leonard Kritharides, David Brieger, Yeu-Yao Cheng, Vincent Chow, Austin Chin Chwan Ng.

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
