## [Decision Letter · Decision Letter 0]

7 Mar 2022

PONE-D-22-00216Sex Differences in Rates of Permanent Pacemaker Implantation and In-hospital Complications: A Statewide Cohort Study of over 7 million persons from 2009-2018PLOS ONE

Dear Dr. Ng,

Thank you for submitting your manuscript to PLOS ONE. After careful consideration, we feel that it has merit but does not fully meet PLOS ONE’s publication criteria as it currently stands. Therefore, we invite you to submit a revised version of the manuscript that addresses the all the points raised during the review process.

We look forward to receiving your revised manuscript.

Kind regards,

Gianluigi Savarese

Academic Editor

PLOS ONE

Journal Requirements:

Reviewers' comments:

Reviewer's Responses to Questions

**Comments to the Author**

1. Is the manuscript technically sound, and do the data support the conclusions?

Reviewer #1: Yes

Reviewer #2: Yes

Reviewer #3: Yes

2. Has the statistical analysis been performed appropriately and rigorously? 

Reviewer #1: I Don't Know

Reviewer #2: Yes

Reviewer #3: Yes

3. Have the authors made all data underlying the findings in their manuscript fully available?

Reviewer #1: Yes

Reviewer #2: No

Reviewer #3: No

4. Is the manuscript presented in an intelligible fashion and written in standard English?

Reviewer #1: Yes

Reviewer #2: Yes

Reviewer #3: Yes

5. Review Comments to the Author

Reviewer #1: The current study reports a gender difference in pacemaker implantation rate at the population level. An assessment of the incidence of intra-operative complications was performed for the new implants. Given the retrospective nature and the lack of etiological hypotheses and data on possible confounders to adjust with, the current study does not allow us to advance hypotheses nor reliable conclusions on the causes of the observed difference.

Despite the above, I consider the study worthy of publication given its application to an unselected population and as an indicator of a difference that requires further investigation and etiological characterization.

Major comments:

- The definition of de novo PPI should be better clarified. Is a generator replacement, if not recurrent during the index admission, included in de novo PPI? Would it be possible to differentiate new pacing system implantation from generator replacement?

I would include PPI implantation rate analysis to de novo implants only as was done for complications. (simplification of the cohort and the reduction of further confounders is crucial for studies that aim to generate hypotheses)

Minor comments:

- In the results paragraph of the abstract, lines 37: (±standard-deviation) move after mean PPI rate lines 35

Reviewer #2: In this manuscript, the authors aim to evaluate a potential bias in PM implantations in men vs. women as well as risk of complications in men vs. women. By using an EHR database, they observed that PM are more frequently implanted in men, but that complications are more frequent in women.

Unfortunately, the main limitation of this manuscript is that only patients with a PM implantation were analyzed, instead of patients with an indication for a PM implantation (or a certain rhythm disorder). Therefore, the conclusion of the manuscript cannot be that there is or is not a bias, as the authors cannot determine if this was done in alignment with local guidelines (e.g. men having a higher risk of having rhythm disorders requiring a PM) or against local guidelines (e.g. no PM implantation in women with a rhythm disorder requiring a PM implantation). This severely limits the relevance of the findings.

Reviewer #3: in the present study the authors explored the sex-related disparities in implantation of permanent pacemaker in a large Australian cohort and concluded that women are undertreated with pacing devices, this difference appears to become wider overtime. The topic related to gender and cardiac devices is attractive, the article is well written and conclusions are reasonable. However, there are some issues that need to be addressed:

- the large sample size allows for more extensive adjustments, in particular considering the high number of demographic and clinical variables. I would consider to assess the association between sex and PPI after correction for most of the available parameters, this might reinforce the message of lower use of PPI in women

- similarly, predictors of PPI-related complications in women vs men would be of interest and might be tested in this specific sample

- there are available information about the socio-economic status (instruction, income, etc)

- The lower age at implantation in men vs women should be further discussed, in relation with concomitant diseases and indications.

- I assume that ICD/CRT-P were not considered in the present study, but it should be more clearly stated

6. PLOS authors have the option to publish the peer review history of their article (what does this mean?). If published, this will include your full peer review and any attached files.

Reviewer #1: **Yes: **Paolo Gatti

Reviewer #2: No

Reviewer #3: No

---

## [Author Response · Author response to Decision Letter 0]

29 Mar 2022

Journal Requirements:

1. Response: We have modified the manuscript to meet PLOS ONE’s style requirements.

2. Response: The New South Wales Population and Health Services Research Ethics Committee prohibits the authors from making the minimal data set publicly available. Interested researchers may contact the Ethics coordinator (ethics@cancerinstitute.org.au) to seek permission to access the data; data will then be made available upon request to all interested researchers who receive approval from the New South Wales Population and Health Services Research Ethics Committee. 

We have revised the Data Availability statement in our resubmission to reflect

the above requirement to assess our study data.

Reviewer #1 

Response: Our study cohort intended to include only patients who had a new pacing system implantation and exclude patients who had recurrent procedures, including replacement of generator (which has a different ACHI procedure code to that of a new pacemaker implantation) between 1-January-2009 and 31-December-2018. We have also revised our PPI rate analysis to de novo implants as suggested. 

 The following changes have been made to the manuscript to clarify our definition of de novo PPI, and that the PPI rate analysis confined to de novo implants. In the Statistical analysis section of the revised manuscript, Page 10, line 146 onwards: “All admissions involving NSW residents between 1-January-2009 and 31-December-2018 were initially collected. The study cohort was limited to NSW residents, stratified by sex, and confined to de novo PPI to reduce confounders. For this study, de novo PPI is defined as patients who had a single PPI procedure during index admission (excluding those who had recurrent procedures or generator replacement between 1-January-2009 and 31-December-2018). This cohort was used to determine the incidence rate and temporal trend of PPI procedures.”

 In the Results section of the revised manuscript, Page 11, line 175 onwards: “Between 1-January-2009 and 31-December-2018, there were 28,714 de novo PPI after excluding non-NSW residents (n=414) and recurrent admissions identified as admissions involving PPI (n=7,033) (S1 Fig.).”

 In the Results section under heading Incidence rate and temporal trend of PPI caseload of the revised manuscript, Page 11, line 181 onwards: “During the 10-year study period, the mean (±SD) implantation rate was 2,871±242 cases per-annum. Though the annual volume of PPI during the study period was steady for both sexes, more men had PPI (Fig 1A). The mean total statewide population was 7,487,393±315,505 persons during the study period, with 50% women (n=3,773,756±334,912). The age-adjusted mean annual PPI was 125.5±11.6 per-100,000-men compared to 63.4±14.3 PPI per-100,000-women, P<0.01 (Fig 1B). Total volume of PPI increased exponentially beyond 40-49yo age group in both sexes (S2 Fig.). When stratified by age groups above 50yo, the rate of PPI per-100,000 in men was consistently double that of women throughout the study period, with the disparity less pronounced in age groups below 50yo (S3 and S4 Figs.). From 2009-2017 (excluding 2018 to limit ascertainment bias), the mean annual rate of increase in PPI in men was 0.9±3.3% compared to 0.4±4.4% in women (P<0.01), suggesting a widening disparity.”. 

Corresponding changes were made to the Abstract to reflect the above changes.

Response: Change made in the revised manuscript Abstract as suggested: “…The mean PPI rate (±standard-deviation) and median age…”.

Reviewer #2

Response: We acknowledge that without knowing the prevalent rates of bradyarrhythmias in our statewide population limits our interpretation of whether a true gender bias in PPM implantations exist or not. As per the discussion part of the manuscript, we reviewed existing literature that had looked at the prevalence of conduction disease in the general population to see if it could partially account for the gender differences in PPM implantation rates seen in the present study. We found a lack of large population studies on the prevalence of bradyarrhythmias that require PPI. By basing our study on a population-wide unselected cohort, we hope our study finding’s nevertheless can serve as a signal that gender differences exist across all age groups, and that it can spur further investigation and etiological characterization.

We have included the following paragraph into the Limitations section on Page 23, line 398 onwards of the revised manuscript: “The study is also limited by unknown rates of bradyarrhythmias and conduction disease in our statewide population and this limits our interpretation of whether a true gender bias in PPI exists or driven by differential rates of conduction disease between gender that requires PPI.”

Reviewer #3

1. Response: We acknowledge the Reviewer comments that it would desirable to adjust for baseline demographic and clinical variables of our study cohort against that of the denominator statewide population. However, we were only able to characterize our statewide population to sex and age, and adjusted these characteristics to reflect the higher rates of PPI in men compared to women across all age groups. It would not be possible to adjust for other demographic or clinical variables of the statewide population as these parameters are not accessible to us.

 However, by virtue of our large sample size, we were able to stratify our study cohort into corresponding specific age groups by decade-year of our statewide population for each calendar year of our study period, and showed that the age-adjusted PPI rates were consistently lower for women compared to men.

2. Response: As suggested, we have now included further analysis on predictors of PPI-related complications between women and men in the revised manuscript, and the additional results are included in the revised Supplementary Material file as Tables S7-S12. 

The following paragraph has been added to the revised manuscript’s result section on Page 18, line 266 onwards: “We further investigated the predictors of in-hospital non-fatal complications in women and separately for men (S7-12 Tables). Similar independent predictors for increased in-hospital complications risk in both men and women were younger age, cardiac valve surgery including TAVI, valvular heart disease, atrial fibrillation and malignancy, while referred as an elective procedure was associated with decreased risk of complications for both genders. Independent predictors associated with increased risk for complications in women but not in men include complete heart block presentation, history of peripheral vascular disease and hypertension. In comparison, these characteristics were associated with increased risk for complications in men but not observed in women: concomitant CABG during PPI admission, history of ischaemic heart disease, congestive cardiac failure, stroke, chronic pulmonary disease and kidney disease.” 

3. Response: Unfortunately, our APDC dataset does not contain socio-economic variables such as income or education level. However, the dataset does contain marital status and country of birth, though we have not included these data variable as we did not consider these to be of clinical relevance to the implantation of pacemakers or its complications. However, should the Reviewer feel otherwise, we will be happy to include these variables into Table 1.

4. Response: The following paragraph has been added into the Discussion section under Sex differences in PPI utilisation heading, Page 19, Line 299 onwards in the revised manuscript: “The median age of women were about 2 years older at time of PPI compared to men in this study. Sick sinus syndrome occurs primarily in older adults,[15] and we noted women had a higher prevalence of sick sinus syndrome as an indication for PPI, which could partially account for the observation of women receiving a pacemaker at an older age. In addition, men are known to have a lower life expectancy (5 years earlier) compared to women,[2] and as shown in this present study, they also have higher prevalence of ischaemic heart disease, complete heart block and other AV block and bradycardia arrhythmias. The combination of lower life expectancy and multiple cardiovascular comorbidities could account for men receiving PPI at a younger age.”

5. Response: Yes, ICD/CRT-P were not considered in this study. 

 We have added the following statement for clarity in the revised manuscript under Methods section, Study population heading, Page 4, line 82: “Patients who had implantable cardioverter defibrillator or cardiac resynchronization therapy pacemaker were not considered for the purpose of this study.”

---

## [Decision Letter · Decision Letter 1]

26 Apr 2022

PONE-D-22-00216R1Sex Differences in Rates of Permanent Pacemaker Implantation and In-hospital Complications: A Statewide Cohort Study of over 7 million persons from 2009-2018PLOS ONE

Dear Dr. Ng,

Thank you for submitting your manuscript to PLOS ONE. After careful consideration, we feel that it has merit but does not fully meet PLOS ONE’s publication criteria as it currently stands. Therefore, we invite you to submit a revised version of the manuscript that addresses the points raised during the review process.

We look forward to receiving your revised manuscript.

Kind regards,

Gianluigi Savarese

Academic Editor

PLOS ONE

Journal Requirements:

Reviewers' comments:

Reviewer's Responses to Questions

**Comments to the Author**

1. If the authors have adequately addressed your comments raised in a previous round of review and you feel that this manuscript is now acceptable for publication, you may indicate that here to bypass the “Comments to the Author” section, enter your conflict of interest statement in the “Confidential to Editor” section, and submit your "Accept" recommendation.

Reviewer #1: All comments have been addressed

Reviewer #3: All comments have been addressed

2. Is the manuscript technically sound, and do the data support the conclusions?

Reviewer #1: Yes

Reviewer #3: Yes

3. Has the statistical analysis been performed appropriately and rigorously? 

Reviewer #1: Yes

Reviewer #3: Yes

4. Have the authors made all data underlying the findings in their manuscript fully available?

Reviewer #1: Yes

Reviewer #3: Yes

5. Is the manuscript presented in an intelligible fashion and written in standard English?

Reviewer #1: Yes

Reviewer #3: Yes

6. Review Comments to the Author

Reviewer #1: The manuscript is well written and easy to follow. The authors have addressed the reviewers' comments. Clarifications were made and analyses added when feasible.

No further comments.

Reviewer #3: The authors have to be commended for having addressed all the concerns in a fashionable and reliable way. However, the lack of information on confounders potentially affecting the relation between gender and PPI remains a major limitation. I understand that it is related to unavailable data, but it has at least to strongly stressed in the limitation section.

7. PLOS authors have the option to publish the peer review history of their article (what does this mean?). If published, this will include your full peer review and any attached files.

Reviewer #1: **Yes: **Paolo Gatti

Reviewer #3: No

---

## [Author Response · Author response to Decision Letter 1]

12 May 2022

We acknowledge the Reviewer’s concern regarding potential confounders as a major limitation. For example, the lack of information on sex-based prevalence rates of bradyarrhythmia and conduction disease at a population-level meant that we cannot exclude this variable as a potential explanation for the sex differences in PPI rates observed in the present study, which we have highlighted as an important limitation of this study. Moreover, we found scarce data available in the literature reporting on the prevalence of bradyarrhythmia at a population-level, highlighting an important area for future research. We also highlighted other important confounders not available to us that may help direct future studies, including clinical data that could potentially affect outcomes of patients receiving a PPI including body mass index, prophylactic antibiotics or anticoagulation usage, and proceduralist experience.

For the reference list, we have reviewed the list twice and identified one dead link. We have updated that link that corresponds to the current literature.

---

## [Decision Letter · Decision Letter 2]

18 Jul 2022

Sex Differences in Rates of Permanent Pacemaker Implantation and In-hospital Complications: A Statewide Cohort Study of over 7 million persons from 2009-2018

PONE-D-22-00216R2

Dear Dr. Ng,

We’re pleased to inform you that your manuscript has been judged scientifically suitable for publication and will be formally accepted for publication once it meets all outstanding technical requirements.

Kind regards,

Yoshihiro Fukumoto

Academic Editor

PLOS ONE

Additional Editor Comments (optional):

Reviewers' comments:

Reviewer's Responses to Questions

**Comments to the Author**

1. If the authors have adequately addressed your comments raised in a previous round of review and you feel that this manuscript is now acceptable for publication, you may indicate that here to bypass the “Comments to the Author” section, enter your conflict of interest statement in the “Confidential to Editor” section, and submit your "Accept" recommendation.

Reviewer #2: All comments have been addressed

Reviewer #3: All comments have been addressed

2. Is the manuscript technically sound, and do the data support the conclusions?

Reviewer #2: Yes

Reviewer #3: Yes

3. Has the statistical analysis been performed appropriately and rigorously? 

Reviewer #2: Yes

Reviewer #3: Yes

4. Have the authors made all data underlying the findings in their manuscript fully available?

Reviewer #2: Yes

Reviewer #3: Yes

5. Is the manuscript presented in an intelligible fashion and written in standard English?

Reviewer #2: Yes

Reviewer #3: Yes

6. Review Comments to the Author

Reviewer #2: The manuscript has significantly improved throughout the review process, its limitations are clearly addressed in the limitation section and the results might be of interest for a broad cardiology readership.

Reviewer #3: All the comments have been addressed, I have no additional concerns

7. PLOS authors have the option to publish the peer review history of their article (what does this mean?). If published, this will include your full peer review and any attached files.

Reviewer #2: No

Reviewer #3: No

---

## [Editor Report · Acceptance letter]

21 Jul 2022

PONE-D-22-00216R2 

Sex Differences in Rates of Permanent Pacemaker Implantation and In-hospital Complications: A Statewide Cohort Study of over 7 million persons from 2009-2018 

Dear Dr. Ng:

I'm pleased to inform you that your manuscript has been deemed suitable for publication in PLOS ONE. Congratulations! Your manuscript is now with our production department. 

Kind regards, 

on behalf of

Dr. Yoshihiro Fukumoto 

Academic Editor

PLOS ONE